# Effect of Revascularization on Intramuscular Vascular Endothelial Growth Factor Levels in Peripheral Arterial Disease

**DOI:** 10.3390/biomedicines10020471

**Published:** 2022-02-17

**Authors:** Larissa Schawe, Ben Raude, Jan Christoph Carstens, Irene Hinterseher, Raphael Donatus Hein, Safwan Omran, Gilles Berger, Nina A. Hering, Matthias Buerger, Andreas Greiner, Jan Paul Frese

**Affiliations:** 1Department of Vascular Surgery, Charité—Universitätsmedizin Berlin, Corporate Member of Freie Universität Berlin and Humboldt-Universität zu Berlin, Hindenburgdamm 30, 12203 Berlin, Germany; ben-heinrich.raude@charite.de (B.R.); jan.carstens@charite.de (J.C.C.); irene.hinterseher@mhb-fontane.de (I.H.); safwan.omran@charite.de (S.O.); matthias.buerger@charite.de (M.B.); andreas.greiner@charite.de (A.G.); jan-paul-bernhard.frese@charite.de (J.P.F.); 2Department of Vascular Surgery, Medizinische Hochschule Brandenburg Theodor Fontane, Ruppiner Kliniken—University Hospital, 16816 Neuruppin, Germany; 3Department of Anaesthesiology and Intensive Care Medicine, Charité—Universitätsmedizin Berlin, Corporate Member of Freie Universität Berlin and Humboldt-Universität zu Berlin, Charitéplatz 1, 10117 Berlin, Germany; raphael-donatus.hein@charite.de; 4Microbiology, Bioorganic & Macromolecular Chemistry, Faculté de Pharmacie, Université Libre de Bruxelles (ULB), Bd du Triomphe, 1050 Brussels, Belgium; gilles.berger@ulb.be; 5Department of General and Visceral Surgery, Campus Benjamin Franklin, Charité—Universitätsmedizin Berlin, Corporate Member of Freie Universität Berlin and Humboldt-Universität zu Berlin, Hindenburgdamm 30, 12203 Berlin, Germany; nina.hering@charite.de

**Keywords:** vascular endothelial growth factor, peripheral artery disease, angiogenesis, revascularization

## Abstract

Vascular endothelial growth factor (VEGF) is a potent driver of angiogenesis, which may help to relieve ischemia in peripheral arterial disease (PAD). We aimed to investigate the role of intramuscular VEGF in ischemic and non-ischemic skeletal muscle in PAD patients before and after surgical or endovascular revascularization and different stages of PAD. Biopsies of the gastrocnemius and vastus muscles from twenty PAD patients with stenosis or occlusion of the superficial femoral artery were obtained both during revascularization and 8 weeks postoperatively. The gastrocnemius muscle was considered ischemic, while vastus muscle biopsies served as intraindividual controls. The levels of vascular endothelial growth factor in muscle lysates were then determined by ELISA. Preoperative VEGF levels were significantly higher in ischemic muscles compared to the controls (98.07 ± 61.96 pg/mL vs. 55.50 ± 27.33 pg/mL, *p* = 0.004). Postoperative values decreased significantly (*p* = 0.010) to 54.83 ± 49.60 pg/mL in gastrocnemius biopsies. No significant change was observed in vastus muscle biopsies, with mean postoperative VEGF values found at 54.16 ± 40.66 pg/mL. Since all patients still had indications for revascularization, impairment of angiogenesis mechanisms can be assumed. More research about angiogenesis in PAD is needed with the ultimate goal to improve conservative treatment.

## 1. Introduction

More than 200 million people worldwide are affected by peripheral arterial disease (PAD) most commonly caused by systemic atherosclerosis. Primary risk factors comprise smoking, diabetes, dyslipidemia, hypertension, obesity, and age [1]. Since most people with PAD are asymptomatic, the disease often remains untreated and undiagnosed in the early stage [2]. Clinical signs include the absence of peripheral pulses and a reduced ankle-brachial index (ABI) [3].

Symptomatic PAD can be subdivided into two groups according to severity: intermittent claudication (IC) and chronic limb-threatening ischemia (CLTI). IC is an aching pain in the affected muscle (calf, thigh, and buttock) that is exacerbated by exercise and relieved by rest. The usual presentation is a set distance of pain-free walking before the onset of claudication. CLTI occurs in advanced stages of PAD, in which perfusion of the affected extremity is depleted to the extent that skin integrity or wound healing is severely compromised. Patients with CLTI experience pain at rest or tissue gangrene [1,4].

The ten-year cardiovascular mortality in asymptomatic PAD patients with ABI < 0.90 is around four times higher than in individuals with normal ABI. In IC, 10-year all-cause mortality is 18% [5]. In CLTI, 5-year mortality is 50%, with a 30% risk of limb amputation at 1-year postdiagnosis [1]. Nonsurgical, conservative treatment options are limited and include lifestyle changes such as smoking cessation, cholesterol reduction, blood pressure control, and antiplatelet therapy [6]. In IC, supervised exercise training may relieve symptoms [7] by developing collateral vessels by angiogenesis and arteriogenesis [8]. Surgical treatment is then indicated if, despite exercise training, the quality of life is reduced due to claudication [9]. For the treatment of CLTI, endovascular or open surgical revascularization is required, as physiological arteriogenesis and angiogenesis are not sufficient [10,11]. Arteriogenesis describes arterial expansion by remodeling existing collateral vessels. It is induced by fluid shear stress and represents macrovascular repair, which is most relevant for tissue perfusion. Angiogenesis refers to the development of blood vessels from pre-existing vasculature, in opposition to vasculogenesis, during which new blood vessels are formed. Angiogenesis is a physiological repair mechanism on the microvascular level [12]. 

There are various proangiogenic factors, such as angiopoietins, VEGF, platelet derived growth factor (PDGF), and hepatocyte growth factor (HGF), as well as several inhibitors, such as interferon-α or interleukin-12. Conditions in which activators exceed inhibitors—a phenomenon called “the angiogenic switch”—include low pO_2_ or pH, hypoglycemia, mechanical stress, injury, immune stimuli, or tumors. PAD results in peripheral tissue hypoxia, which triggers the transcription of hypoxia-inducible factor 1 (HIF) and is a major inducer of angiogenesis. HIF is responsible for the upregulation of several genes and VEGF is among them. 

Under the term VEGF, several family members are subsumed, namely VEGF-A through -D, and several subtypes of VEGF-A bind to three receptors: VEGF-1 to VEGF-3. VEGF-A binds to VEGFR-1 and -2. VEGFR-1 is expressed on endothelial cells, smooth muscle cells, and vascular and hematopoietic stem cells. By activation of VEGFR-2, VEGF-A is mainly active on the vascular endothelium, stimulating endothelial proliferation, migration, and vascular permeability. VEGF-A mediates almost all phases of angiogenesis, from initial vasodilatation to permeability of vascular endothelium and migration and proliferation of endothelial cells [13].

Several studies aimed to increase local angiogenic factors to expand conservative treatment options in PAD [10,11,14]. However, these strategies using proangiogenic factors such as VEGF, HGF, and fibroblast growth factor (FGF) did not appear to reduce the rate of limb amputation or death [14].

In the present study, we acquired muscle biopsies and measured VEGF levels before and after surgical revascularization to gain insights into disease mechanisms and angiogenesis in PAD. We hypothesized that VEGF increased in PAD patients’ ischemic muscles compared to their non-ischemic muscles. We surmised that, after revascularization, VEGF levels in the revascularized muscle would decrease from preoperative levels.

## 2. Materials and Methods

Twenty adults suffering from symptomatic PAD were included in this prospective, nonrandomized controlled study. The study was approved by the local ethics committee under proposal number EA4/077/19.

Inclusion criteria were severe intermittent claudication (Fontaine stage IIb, claudication after less than 200 m walking distance) or chronic critical limb-threatening ischemia (CLTI) caused by occlusion or stenosis of the superficial femoral artery (SFA). In some patients, additional lesions distal of the SFA were present. Exclusion criteria were acute limb ischemia; significant stenosis or occlusion of the aortic bifurcation, of the iliac arteries, common femoral artery, deep femoral artery, or lateral circumflex femoral artery; active malignancy; pregnancy; or inability to provide informed consent. Samples were collected from February 2016 to August 2021. 

Revascularization of the SFA was performed by endarterectomy, percutaneous transluminal angioplasty (PTA), a combination of those (hybrid surgery), or bypass surgery. Muscle biopsies of the gastrocnemius and vastus muscle were obtained during revascularization surgery. Eight weeks after operative treatment, muscle biopsies were taken during follow-up in our outpatient department under local anesthesia. Ankle-brachial index (ABI) was measured before surgical treatment as well as eight weeks postoperatively. The success of revascularization was confirmed by clinical improvement of the patient, ABI improvement, and duplex ultrasound. 

As all included patients suffered from an occlusion or stenosis of the SFA, the gastrocnemius muscle was considered ischemic, and the vastus muscle of the same patient was the non-ischemic control, respectively. Perfusion of the vastus muscle was provided by deep femoral and lateral circumflex femoral arteries, which were proven to be patent in our patients. The technique of muscle sampling using a Bergstrom needle has been described by Gratl et al. [15]. Furthermore, they provided evidence that the gastrocnemius muscle can in fact be regarded as ischemic, whereas the vastus muscle can be considered healthy in comparison to healthy controls [15].

Samples were immediately frozen in liquid nitrogen and stored at −80 °C. We used CelLytic MT tissue lysis/extraction reagent (Sigma-Aldrich, Inc., Saint Louis, MI, USA. Product Code C 3228) to lyse cells and release protein and RNA contents. The QuantiPro™ BCA Assay Kit (Sigma-Aldrich) was used to identify the protein content. We used a sandwich enzyme ELISA (Enzyme-linked Immunoabsorbent Assay) technique (Quantikine Human VEGF Immunoassay, R&D systems, Inc., Minneapolis, MN, USA) to determine VEGF levels in skeletal muscle lysates, as per manufacturer’s protocol. SPSS (IBM Corp. Released 2017. IBM SPSS Statistics for MacOS, Version 25.0., Armonk, NY, USA) was used for statistical analysis. Most continuous variables were not normally distributed; therefore, the results were presented as median and interquartile range (IQR). Group comparisons of preoperative versus postoperative values and ischemic versus non-ischemic muscles were performed using a Wilcoxon test for paired samples. For continuous variables, correlations were analyzed using bivariate Pearson’s correlation. In order to compare the characteristics of different groups, a Mann–Whitney U Test was conducted. A *p*-value < 0.05 was considered significant. 

## 3. Results

Twenty individual patients suffering from PAD were included in this study. Table 1 shows clinical characteristics, cardiovascular risk factors, and the technique of revascularization. In all 20 patients, at least one vascular risk factor was prevalent. Four patients had diabetes (20%). The ABI was 0.66 ± 0.26 before surgery and improved significantly to 0.97 ± 0.23 after revascularization (*p* = 0.036).

The two groups of IC and CLTI were not different (*p* > 0.05) regarding age, BMI, ABI, and cardiovascular risk factors. 

**Table 1 biomedicines-10-00471-t001:** Patient characteristics, cardiovascular risk factors, and surgical techniques. IQR: interquartile range. PAD: peripheral artery disease. IC: intermittent claudication. CLTI: chronic limb-threatening ischemia.

*n* = 20		*n* (%)	Median	IQR
Male gender		18 (90)		
Female gender		2 (10)		
Severity of PAD	IC	13 (65)		
	CLTI	7 (35)		
Diabetes		4 (20)		
Hypertension		13 (65)		
Active smoking		11 (55)		
Coronary artery heart disease		6 (30)		
Dyslipoproteinemia		11 (55)		
Chronic kidney disease (CKD)		6 (30)		
Age (years)			66.05	11.90
Body mass index (BMI)			25.30	7.40
Ankle brachial index (ABI) before revascularization			0.66	0.26
ABI after revascularization			0.97	0.23
Technique of revascularization	Open	9 (45)		
	Endovascular	2 (10)		
	Hybrid	9 (45)		

### VEGF Results

In ischemic gastrocnemius muscles, preoperative VEGF levels were 98.07 ± 61.96 pg/mL. Eight weeks after successful revascularization, the muscular VEGF content decreased significantly to 54.83 ± 49.60 pg/mL (*p* = 0.010). The VEGF levels in non-ischemic vastus muscles did not change throughout revascularization (pre-op. 55.50 ± 27.33 pg/mL vs. post-op. 54.16 ± 40.66 pg/mL) (Table 2, Figure 1). The median difference in preoperative to postoperative values was 2.78 ± 32.18 pg/mL in vastus biopsies, compared to 34.29 ± 47.72 pg/mL in gastrocnemius biopsies (*p* = 0.02). 

Looking at IC only (Figure 2), VEGF levels dropped from 123.56 ± 62.91 pg/mL preoperatively to 55.89 ± 48.03 (*p* = 0.046) after revascularization in ischemic muscles. Comparing gastrocnemius muscle to vastus muscle, we observed significantly higher VEGF levels in the ischemic gastrocnemius muscles than vastus muscles (123.56 ± 62.91 pg/mL vs. 57.84 ± 26.12 pg/mL; *p* = 0.011). 

In CLTI, VEGF levels of gastrocnemius muscle tissue decreased from 76.83 ± 44.76 pg/mL before revascularization to 54.37 ± 35.62 pg/mL postoperatively, *p* = 0.091. The difference of VEGF levels in ischemic vs. non-ischemic muscles was not significant in the CLTI group (76.83 ± 44.76 pg/mL vs. 53.16 ± 18.49 pg/mL, respectively, *p* = 0.176).

In the subgroup of four patients (20%) with diabetes (Figure 3), preoperative VEGF values in ischemic muscle tissue were 74.4 ± 30.36 pg/mL, compared to 101.18 ± 63.93 pg/mL in nondiabetics. Postoperatively, these VEGF levels decreased to 24.33 ± 30.73 pg/mL in diabetics (*p* = 0.273) and 55.59 ± 45.11 in nondiabetics (*p* = 0.030). The preoperative VEGF levels in vastus muscle tissue were 75.01 ± 47.16 pg/mL in patients with diabetes, showing similar values to preoperative ischemic muscle tissue. In contrast, we found 54.44 ± 27.51 pg/mL VEGF levels in nondiabetic vastus biopsies, showing a significant difference relative to the VEGF levels of ischemic nondiabetic muscle tissues (*p* = 0.001). 

## 4. Discussion

Our study aimed to determine the intramuscular levels of proangiogenic growth factor VEGF in different stages of PAD and to investigate the effects of surgical revascularization. We have found elevated levels of proangiogenic growth factor VEGF in ischemic muscle biopsies compared to non-ischemic controls. By investigating the effect of surgical revascularization, we have found a significant decrease in VEGF levels postoperatively. Patients with advanced stages of PAD or with diabetes showed lower VEGF levels in ischemic muscles. A detailed understanding of the roles of growth factors in PAD and the possible dysfunction of angiogenesis and arteriogenesis mechanisms is lacking; thus, it became the focus of our research. More precise knowledge about these mechanisms could possibly reform conservative PAD treatments. 

Very few studies reported intramuscular VEGF contents in human skeletal muscles [16,17,18]. The biological response of VEGF is dependent on its local tissue concentration [19]. More studies exist on plasma or serum concentrations, but these are affected by platelet-bound VEGF that was detached by coagulation. It is presumed that muscle VEGF content constitutes close to half of the human body’s VEGF [20].

PAD patients in a real-life population constitute a heterogeneous group. By including patients with a pathology of SFA, the group’s pathophysiological homogeneity was ensured, and the gastrocnemius muscle was defined as ischemic. The vastus lateralis muscle was considered non-ischemic and could be used as an intraindividual control [21]. Therefore, no separate control group of healthy individuals undergoing invasive muscle sampling was included. In all patients included in this study, at least one cardiovascular risk factor was prevalent. No correlation was found between VEGF levels and cardiovascular risk factors, ankle-brachial index, or age. One study shows that systemic VEGF levels are higher with a rising BMI [22], including all stages of PAD and all locations and types of atherosclerotic lesions. In our study in isolated SFA pathologies and subgroups of IC and CLTI, we could not find a significant correlation between BMI and intramuscular VEGF. 

VEGF levels were significantly higher in ischemic muscle tissues than non-ischemic muscle tissues. This supports the results of Jalkanen 2008 [23] and Findley 2008 [24], showing elevated VEGF in the plasma of PAD patients. In contrast to Jalkanen 2008, our VEGF muscle content was higher in IC and not in CLTI. The relatively lower level in CLTI compared to IC could be an effect of detraining in our CLTI patients, as their lifestyle is more sedentary, caused by resting pain and tissue loss. Exercise is a significant inducer of VEGF [16].

Furthermore, elevated VEGF levels could be seen as part of a compensatory mechanism to induce angiogenesis and, thus, overcome hypoxia. The lower VEGF in CLTI than in IC could represent an “exhaustion” or an impairment of the angiogenesis system due to chronic critical hypoxia [25]. It can be assumed that VEGF levels, even though significantly elevated, were either insufficient to form collaterals or that the downstream response to elevated VEGF was impaired. Since all patients enrolled in this study had indications for revascularization according to guidelines [9], the compensatory VEGF response was not sufficient to relieve their symptoms. Furthermore, proangiogenic growth factor injections as possible treatment for PAD have not yet delivered convincing clinical results [14].

In diabetic patients, we could not find an elevation of VEGF levels in the ischemic muscle, confirming the results of Wieczor et al. [26]. Thus, a potential regulatory mechanism was absent in diabetics. Angiogenesis in diabetic PAD patients is impaired due to hyperglycemia [26,27]. In contrast, hypoglycemia is an inducer of the angiogenic switch [13].

Angiogenesis, in addition to its induction as a therapeutic strategy, has garnered much attention in the past two decades. The pathologic elevation of VEGF levels in pretherapeutic PAD could be part of a physiological compensatory mechanism to overcome the resulting peripheral hypoxia [19]. An upregulation to compensate for hypoxia in PAD as a multisystem disease was also found in mitochondrial respiratory chain abnormalities [15]. However, the formation of new blood vessels also constitutes a hallmark of cancer and sustains tumor growth. The physiological response to upregulation of VEGF seems to be limited. Supraphysiological levels of VEGF result in aberrant and lacunar vessel growth [19], as observed in various entities of cancer. In solid malignancies, VEGF serum concentrations in the range of 200 to 700 pg/mL could be found [28,29,30]. In the present study, the highest VEGF concentration was 190.6 pg/mL, with average levels below 100 pg/mL. 

After successful revascularization confirmed by increased ABI, we found a significant decrease in VEGF levels in the previously ischemic gastrocnemius muscle tissue. No change by revascularization was found in the non-ischemic vastus muscle. Postoperatively, the difference of ischemic to non-ischemic muscle had disappeared. This may indicate that intramuscular VEGF levels are regulated locally and exhibit local effects. To our knowledge, this is the first study on muscular VEGF levels before and after surgical interventions. There is one study regarding plasma vascular growth factors in a surgical cohort. Yoshitomi et al. demonstrated a decrease in HGF serum concentrations after peripheral bypass surgery or percutaneous transluminal angioplasty (PTA) [31]. Muscular VEGF levels in nonsurgical patients were assessed by Jones et al. [18] before and after supervised exercise training (SET). In an ischemic gastrocnemius muscle, SET resulted in a decrease in VEGF content.

Interestingly, their pretherapeutic VEGF levels were the same as in healthy controls. SET patients usually have a longer pain-free walking distance and a lesser degree of PAD than PAD patients with an indication for surgery. Surgical treatment is indicated in higher Rutherford stages and when conservative management is exhausted. 

Limitations of this study are the small sample size and the lack of an appropriately sized control group of healthy individuals. By including only SFA lesions and taking the non-ischemic vastus muscle as control, interindividual variability was eliminated, and invasive muscle sampling of healthy subjects was not justified. 

Recently, interest is rising with respect to investigating different angiogenic and nonangiogenic subgroups of VEGF. In a murine model, Kikuchi et al. [32] demonstrated that the antiangiogenic isoform of VEGF, VEGF_165b_, is elevated in PAD. This could be reproduced in human PAD patients by Ganta et al. [33], who showed elevated VEGF_165b_ isoforms and decreased levels of pro-angiogenic VEGF_165a_ in PAD patients. Future research is needed to investigate the differential effects of VEGF isoforms and other factors involved in angiogenesis. This could help in better understanding the process and can reveal new strategies in the treatment of PAD, especially in patients who are unfit or ineligible for vascular surgery. 

## 5. Conclusions

In our study, muscular VEGF content in the ischemic skeletal muscle was elevated in pretherapeutic PAD patients. Successful surgical revascularization resulted in a decrease in VEGF to a level similar to non-ischemic muscles. 

Physiological angiogenesis would be expected considering the pretreatment significant elevation of proangiogenic growth factor VEGF. Nevertheless, the patients included had a clear indication for surgical treatment according to current guidelines. An insufficiency of angiogenesis and arteriogenesis in PAD may be a possible explanation. Our results show a more pronounced elevation of VEGF levels in IC and in nondiabetic patients, whereas in advanced stages of PAD and in diabetics, we could not show a significant increase in VEGF levels. It is possible that angiogenesis and arteriogenesis mechanisms are more severely impaired in more critical stages of PAD and diabetics.

Further work is necessary to understand arteriogenesis and angiogenesis in PAD, especially focusing on the proangiogenic and antiangiogenic isoforms of VEGF.

## Figures and Tables

**Figure 1 biomedicines-10-00471-f001:**
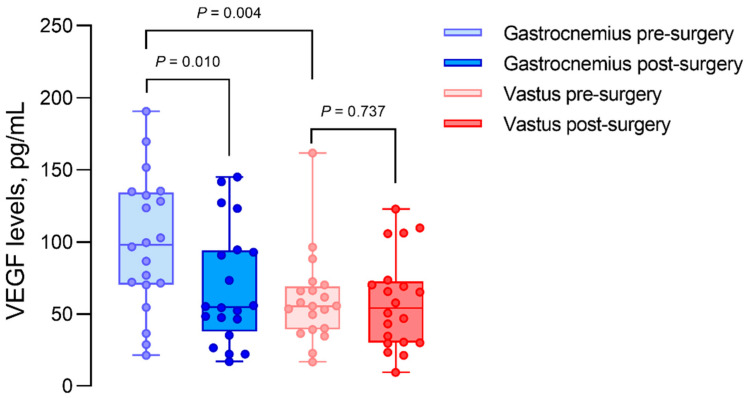
Intramuscular VEGF concentrations in gastrocnemius and vastus lateralis muscle biopsies, before (pre) and after (post) revascularization surgery. Gastrocnemius muscles are considered ischemic, and vastus muscles are considered as non-ischemic controls.

**Figure 2 biomedicines-10-00471-f002:**
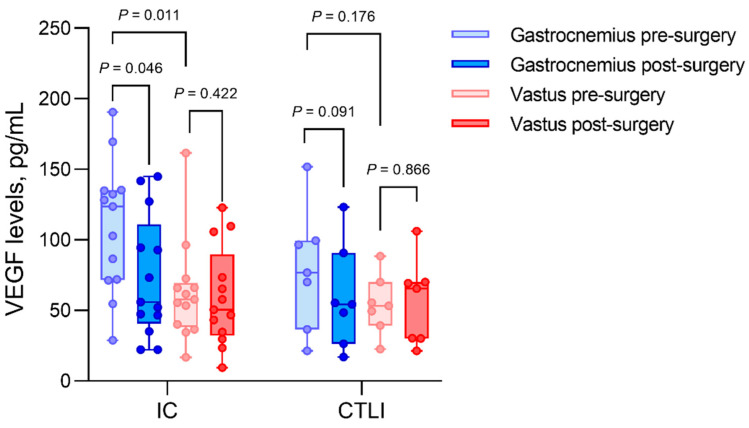
Intramuscular VEGF concentrations of patients with intermittent claudication (IC) compared to critical limb ischemia (CLTI) in gastrocnemius and vastus lateralis muscle biopsies, before (pre) and after (post) revascularization. Gastrocnemius muscles are considered ischemic, and vastus muscles are considered as non-ischemic controls.

**Figure 3 biomedicines-10-00471-f003:**
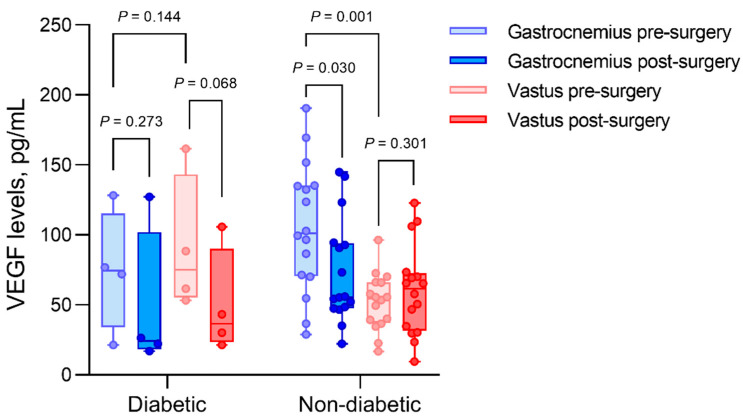
Intramuscular VEGF concentrations of patients with diabetes compared to nondiabetic patients in gastrocnemius and vastus lateralis muscle biopsies, before (pre) and after (post) revascularization. Gastrocnemius muscles are considered ischemic, and vastus muscles are considered as non-ischemic controls. VEGF levels were not different when comparing hypertensive to non-hypertensive patients, as for dyslipidemia, sex, smoking status, and chronic kidney disease. VEGF levels were not correlated to BMI, age, or ABI.

**Table 2 biomedicines-10-00471-t002:** VEGF levels in biopsies of vastus lateralis muscles (non-ischemic) and gastrocnemius muscles (ischemic) before and after revascularization. *p*-values were calculated using a Wilcoxon test for paired samples. An asterisk indicates *p* < 0.05. IQR: interquartile range. PAD: peripheral artery disease. IC: intermittent claudication. CLTI: chronic limb-threatening ischemia.

VEGF (pg/mL)	Preoperative		Postoperative		*p*-Value
	Median	IQR	Median	IQR	
Gastrocnemius all patients	98.07	61.96	54.83	49.60	0.010 *
-Gastrocnemius IC	123.56	62.91	55.89	48.03	0.046 *
-Gastrocnemius CLTI	76.83	44.76	54.37	35.62	0.091
-Gastrocnemius diabetics	74.40	30.36	24.33	30.73	0.273
-Gastrocnemius nondiabetics	101.18	63.93	55.59	45.11	0.030 *
Vastus all patients	55.50	27.33	54.16	40.66	0.737
-Vastus IC	57.84	26.12	50.60	38.85	0.422
-Vastus CLTI	53.16	18.49	65.60	39.45	0.866
-Vastus diabetics	75.01	47.16	36.62	30.91	0.068
-Vastus nondiabetics	54.44	27.51	61.47	37.41	0.301

## Data Availability

Data are contained within the article. The raw data presented in this study are available upon request from the corresponding author.

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
