# Peer review of "Effect of Revascularization on Intramuscular Vascular Endothelial Growth Factor Levels in Peripheral Arterial Disease"

_biomedicines, 2022, doi:10.3390/biomedicines10020471_

Round 1

Reviewer 1 Report

Authors investigated intramuscular levels of VEGF in PAD patients, which helped gain some insights into the role of physiological angiogenesis, regulatory/compensatory mechanisms, and the effect of revascularization or exercises. Despite the significance of the study (the first study on pre- and post-operative VEGF levels and the findings presented in the manuscript), this reviewer has major and minor concerns as follows.

Major concerns:

  1. Although the limitation of this study (lack of normal controls) was explained, it is very hard to say that 'VEGF alone does not induce sufficient angiogenesis in PAD.' (lines 264-265)
  2. Authors should examine the levels of typical isoforms of VEGF-A, including VEGF165b.
  3. Authors should mention other members of VEGF family such as VEGF-B, -C, and -D.
  4. Male vs. Female difference (or similarity) must be provided and interpreted. All figures or tables must have female data and noted in different color or symbols.
  5. Objective evidence must be provided that the gastrocnemius musucles were ischemic and the vastus muscle of the same patient non-ischemic.

Minor conerns:

  1. More figures can be provided for easier reading and interpretation of Table 2. Moreover, a figure on exhaustion or an impairment of angiogenesis system can be provided with current working model.
  2. The phrase such as 'before revascularizaiton or pre-operative' can be added for clarity in lines 157, 162, and 170.
  3. Line 142 and line 171 : check the p values which differ between the two.
  4. More recent references within the last 5 years can be added.
  5. For efficient blood flow, there has to be an arteriogenesis rather than angiogenesis. A paragraph on this issue can be added.

Author Response

We would like to thank you for the prompt response and the beneficial comments. We are grateful for the opportunity to submit a revised version of the manuscript. Please find below a point-by-point statement of the changes we made to the manuscript. In the manuscript, changes are marked in red. We’d be happy to provide an unmarked version as well.

We sincerely hope that we have adequately addressed all of the issues raised. We are looking forward to your feedback.

Major concerns:

Although the limitation of this study (lack of normal controls) was explained, it is very hard to say that 'VEGF alone does not induce sufficient angiogenesis in PAD.' (lines 264-265)

The reviewer is right. We weakened our statement regarding this conclusion of our study, and changed the abstract.

Authors should examine the levels of typical isoforms of VEGF-A, including VEGF165b.

Thank you for bringing this up. Recent publications indicate a possible elevation of the anti-angiogenic isoform in PAD (Ganta et al. Circ Res. 2017; Kikuchi R et al, Nat Med. 2014 )

We think this is very interesting and we are planning to examine VEGF165b and VEGF165a as soon as possible. Unfortunately, the lysates of our muscle biopsies are only enough to perform one ELISA, which is why we are currently collecting more probes for this future project. We are hopeful, that it will bring many new insights.

Authors should mention other members of VEGF family such as VEGF-B, -C, and -D.

We added this information to the Introduction section, page 2 lines 76-83.

Male vs. Female difference (or similarity) must be provided and interpreted. All figures or tables must have female data and noted in different color or symbols.

We agree, that it is most definitely meaningful to examine the possible gender differences. In this study however, we could not find any significant differences in VEGF levels between male and female patients. One limitation of our study is the low number of female patients. In the future, we intend to include more female patients for further investigation.

Objective evidence must be provided that the gastrocnemius musucles were ischemic and the vastus muscle of the same patient non-ischemic.

The inclusion criteria of our study were selected with the intention to include a homogenous cohort of PAD patients with atherosclerotic lesions of the SFA. In several patients, additional lesions more distally were present. PAD was confirmed by a pathologic ankle brachial index and walking impairment in a treadmill test. Patients with flow-limiting pathologies of the aorta, iliac, common or deep femoral arteries were excluded, to ensure that vastus muscles were non-ischemic, and could be used as intra-individual controls.

We sharpened this definition in the methods section (Page 3, lines 101-104, 114-121) There is data that vastus muscle in PAD without aortoiliac pathology may be considered as non-ischemic, in comparison with vastus muscle from healthy controls. (Gratl et al, Eur J Vasc Endovasc Surg. 2020)

Minor conerns:

More figures can be provided for easier reading and interpretation of Table 2. Moreover, a figure on exhaustion or an impairment of angiogenesis system can be provided with current working model.

We added the new figure 2, page 6 depicting VEGF results in IC vs. CLTI.

We also added figure 3, page 6, depicting VEGF results in diabetics vs. non-diabetics.

The phrase such as 'before revascularizaiton or pre-operative' can be added for clarity in lines 157, 162, and 170.

Thank you, we have changed the phrases as suggested.

Line 142 and line 171 : check the p values which differ between the two.

The reviewer is right,  in the footnotes of the table 2 we have stated, that the p-value was p<0.01 and in the text we have given the exact value, which is p=0.001. You will now find the exact p-values in a new figure 3 we added, as well as in the text.

More recent references within the last 5 years can be added.

It is true, that there are only 11 references of the past 5 years, but we started this study in 2016. During preparations we read the literature from 2016 and earlier. With the 11 more recent references we added, we think to have added all the additional information that is relevant for the present study.

For efficient blood flow, there has to be an arteriogenesis rather than angiogenesis. A paragraph on this issue can be added.

We have expanded our paragraph on arteriogenesis and angiogenesis, Introduction section, P2, Lines 62-67)

Reviewer 2 Report

In the current manuscript Schawe and colleagues report on intramuscular VEGF levels in pre- and postrevascularization PAD patients (n=20). They use an internal non-ischemic control which is vastus muscle, which is actually a very nice solution and even more reliable than a true healthy control group, at least this is my opinion. They show that in general the VEGF levels are higher in the ischemic gastrocnemius muscle and levels drop after succesful revascularization. There are some issues that could be improved or require some attention.

Abstract:

  1. The authors conclude that angiogenic response in PAD patients is insufficient. This can not be concluded based on this population. At maximum you could conclude that in a subset of patients with PAD the response is insufficient, but at some point it can also be the case that only an angiogenic response is just not enough to overcome the macrovascular pathology that critically reduces perfusion. The latter probably less likely in this population with isolated SFA pathology.
  2. L24; Isolated SFA stenosis. Also patients with occlusions were included right?
  3. Reading the abstract and also methods the atuhors state surgical revascularization, which is often considered bypass surgery. I would be more explicitly that patients in this study either underwent bypass surgery, endovascular revascularization or hybrid surgery (and specify this) instead of mentioning this surgical revascularization.

Introduction:

4. In the last sentence of the introduction, L81 and further. The authors state "after bypass surgery", but if I am correct not all patients underwent bypass surgery, please revise this passage.

Methods:

5. CLTI patients were also included based on isolated SFA pathology. Did these patients really have only SFA pathology? CLTI is almost never seen in atherosclerotic disease in a single vascular segment. Almost always crural or multilevel pathology. Please specificy and correct if necessary.

Results:

6. Table 1: A bit messy. Text in first column shifts to the right, reads not very easy. Please correct.

7. Only mention the most relevant findings from table 1 in the results section. In the current form it is almost a copy of the table.

8. Were there differences in VEGF levels between endovascular/hybrid and bypass patients? Especially post-operative. Considering the potential effect of surgical trauma and muscle response on VEGF levels.

9. The difference in pre-operative levels between CLTI and IC in VEGF are striking. This could be a reflection of an impaired angiogenic response leading to CLTI in a subpopulation with a similar disease pattern. Please discuss.

10. L160-161: Would mention it a trend here since numbers are not significant, but of course very likely that with a larger sample size this was observed.

Discussion:

11. Start the discussion with the main findings of the study and not with the aim of the study. This is already mentioned at the end of the introduction. So, start discussion with mentioning what kind of important and novel insights it gives and what this could mean for clinic and science.

12. Limitiations: I would also consider the relative small sample a limitation, as well as the heterogeneous population (CLTI and IC, different forms of surgery, diabetics and non-diabetics) which could lead to confounding, and only a single follow-up sample at 8 weeks (understand why, but would be interesting to have for instance an extra sample at 6 months).

Conclusion:

13. Would choose for a more condensed conclusion and try to stick more to the objective data of the study. Actually you found that VEGF decreases after revascularization and is lower in ischemic than in non-ischemic muscle. And probably that it is less pronounced in CLTI than IC, which might suggest a different response in the two groups or a less functional angiogenic response in CLTI.

There are different ways to draw conclusions from the manuscript, but the current version does not really fit the findings.

Author Response

We would like to thank you for the prompt response and the beneficial comments. We are grateful for the opportunity to submit a revised version of the manuscript. Please find below a point-by-point statement of the changes we made to the manuscript. In the manuscript, changes are marked in red. We’d be happy to provide an unmarked version as well.

We sincerely hope that we have adequately addressed all of the issues raised. We are looking forward to your feedback.

The authors conclude that angiogenic response in PAD patients is insufficient. This can not be concluded based on this population. At maximum you could conclude that in a subset of patients with PAD the response is insufficient, but at some point it can also be the case that only an angiogenic response is just not enough to overcome the macrovascular pathology that critically reduces perfusion. The latter probably less likely in this population with isolated SFA pathology.

The reviewer is right. We weakened our statement regarding this conclusion of our study, and changed the discussion section and the abstract.

L24; Isolated SFA stenosis. Also patients with occlusions were included right?

Thank you, you are absolutely correct, we have changed the phrase. (Abstract, Line 23)

Reading the abstract and also methods the atuhors state surgical revascularization, which is often considered bypass surgery. I would be more explicitly that patients in this study either underwent bypass surgery, endovascular revascularization or hybrid surgery (and specify this) instead of mentioning this surgical revascularization.

We have added more detailed description for clarity. The details of the revascularization procedures are provided in table 1.

Introduction:

In the last sentence of the introduction, L81 and further. The authors state "after bypass surgery", but if I am correct not all patients underwent bypass surgery, please revise this passage.

 Thank you for remarking this, we changed the passage.

Methods:

CLTI patients were also included based on isolated SFA pathology. Did these patients really have only SFA pathology? CLTI is almost never seen in atherosclerotic disease in a single vascular segment. Almost always crural or multilevel pathology. Please specificy and correct if necessary.

Thank you for this precise remark. We sharpened the description of the inclusion and exclusion criteria. All patients had flow-limiting stenosis or occlusion of the SFA. In several patients, an additional stenosis or occlusion more distally was present.  Flow-limiting stenosis of the aorta, the iliac arteries, the common femoral artery, the deep femoral artery or the lateral circumflex femoral artery, were excluded from the study, to ensure that the blood flow to the lateral vastus muscle was intact and we could still take it as the non-ischemic control biopsy. A significant improvement of the ABI postoperatively, improvement of clinical symptoms and a significant decrease in VEGF levels however indicated in all patients, that by revascularization of the SFA and/or popliteal artery restored the perfusion blood-flow significantly. We therefore regarded a stenosis/occlusion of the arteries of the calf/foot as less significant for this study, as this would probably not alter the status of the gastrocnemius muscle before and after revascularization

Results:

Table 1: A bit messy. Text in first column shifts to the right, reads not very easy. Please correct.

Our apologies for the format, we have corrected it.

Only mention the most relevant findings from table 1 in the results section. In the current form it is almost a copy of the table.

We shortened the first paragraph of the results section as suggested (P4, lines 154-159)

Were there differences in VEGF levels between endovascular/hybrid and bypass patients? Especially post-operative. Considering the potential effect of surgical trauma and muscle response on VEGF levels.

Thank you for this interesting question. When comparing VEGF levels of different revascularization techniques, we could not see any significant differences. It is to say that only two patients underwent endovascular revascularization only, which is a relatively small group of patients with less surgical trauma. In the future, we intend to include more patients with only endovascular revascularization, in order to investigate this further.

The difference in pre-operative levels between CLTI and IC in VEGF are striking. This could be a reflection of an impaired angiogenic response leading to CLTI in a subpopulation with a similar disease pattern. Please discuss.

This difference is indeed striking and one key finding in our study. Please find the discussion of this in the discussion section (lines 238-254)

L160-161: Would mention it a trend here since numbers are not significant, but of course very likely that with a larger sample size this was observed.

We absolutely agree with you here and we intend to repeat the study with larger samples size of each subgroup, in order to draw conclusions from that. We have added a passage to the discussion where we mention this. (lines 288-289)

Discussion:

Start the discussion with the main findings of the study and not with the aim of the study. This is already mentioned at the end of the introduction. So, start discussion with mentioning what kind of important and novel insights it gives and what this could mean for clinic and science.

Please find an improved first paragraph  of the discussion section in the revisedmanuscript. (lines 211-220)

Limitiations: I would also consider the relative small sample a limitation, as well as the heterogeneous population (CLTI and IC, different forms of surgery, diabetics and non-diabetics) which could lead to confounding, and only a single follow-up sample at 8 weeks (understand why, but would be interesting to have for instance an extra sample at 6 months).

Thank you, this is indeed a limitation of our study. We added this to the discussion section (lines 288-289). Future studies would include a larger cohort to allow subgroup analyses. Regarding additional control biopsies, we followed the protocol of one other study (Gratl et al, Eur J Vasc Endovasc Surg. 2020), indicating a normalization of mitochondrial function after 6-8 weeks, which renders more biopsies ethically questionable. If this is the case also for VEGF is a very interesting question for a future study.

Conclusion:

Would choose for a more condensed conclusion and try to stick more to the objective data of the study. Actually you found that VEGF decreases after revascularization and is lower in ischemic than in non-ischemic muscle. And probably that it is less pronounced in CLTI than IC, which might suggest a different response in the two groups or a less functional angiogenic response in CLTI. There are different ways to draw conclusions from the manuscript, but the current version does not really fit the findings.

Thank you, the reviewer is right. We shortened and sharpened the conclusion, focusing on the objective results (lines 308-315)

Round 2

Reviewer 1 Report

I understand the difficulties authors faced in performing further studies. Other points were addressed adequately except the marking of female data in the accompanying graphs.